# Development of an Assessment of Ethics for Chinese Physical Education Teachers: A Study Using the Delphi and Expert Ranking Methods

**DOI:** 10.3390/ijerph191911905

**Published:** 2022-09-21

**Authors:** Haohui Liu, Zhihua Yin, Sitong Chen, Youcai Yang, Hengxing Tian

**Affiliations:** 1College of Physical Education and Health, East China Normal University, Shanghai 200241, China; 2Sports Postdoctoral Mobile Station, Tsinghua University, Beijing 100084, China; 3Institute for Health and Sport, Victoria University, Melbourne 8001, Australia; 4China Basketball College, Beijing Sports University, Beijing 100084, China

**Keywords:** physical education teacher, ethics, assessment, AECPET, Delphi method, expert ranking method, China

## Abstract

Background: Developing the ethics of physical education (PE) teachers is important for promoting the overall development of students. However, it is unclear which indicators can be used to assess the ethics of PE teachers in China. Therefore, this study aimed to develop an assessment of ethics for Chinese physical education teachers (AECPET) using the Delphi and expert ranking methods. Methods: Two rounds of the Delphi method were performed to develop the assessment. An expert ranking method was used to determine the weight of each domain and indicator. Results: The developed AECPET is a multi-dimensional model with eight domains: (1) Policy Implementation (PI), (2) Legal Compliance and Patriotism (LCP), (3) Love for Students (LS), (4) Daily Performance (DP), (5) Philosophy of Educating Students (PES), (6) Attitude towards Scientific Research (ASR), (7) Awareness of Self-discipline and Honesty (ASH), and (8) Attitude towards Serving Society (ASS); and 42 indicators. The weight of PI, LCP, LS, DP, PES, ASR, ASH, and ASS are 18.1%, 19.1%, 16.2%, 11.1%, 16.3%, 7.2%, 9.1%, and 4.6%, respectively. Conclusions: The AECPET is an evaluating system constructed based on the perspective of Chinese PE teachers, which also has a potentially global perspective and can be used by PE teachers with different cultural views. Applied by the government, schools, PE teachers, and students, the AECPET can improve the level of ethics of Chinese PE teachers.

## 1. Introduction

The definition of teachers’ ethics varies by scholar. Campbell described it as something intangible that should be paid attention to in teaching [1]. Li proposed that it refers to the quality and behavior standards of teachers [2], while Liu defines it as the sum of teachers’ moral character, code of conduct, and ideas formed in specific educational practice [3]. Subsequently, in this study, teachers’ ethics is conceptualized as the level of morality, responsibility, kindness, and the overall demeanor of teachers, and it is regarded as a fundamental standard for evaluating teachers’ performances. As teachers hold the role of moral agents [4], their professional ethics are important to the quality of school education and, consequently, the development of the moral level of society. While teachers with high-quality ethics can help students grow and influence students to develop high morals through their performance, their questionable behavior will negatively impact society and make teachers lose credibility [5]. 

The ethics of Physical Education (PE) teachers is of great importance. In terms of students’ ethics development, Drewe [6] indicated that there was a relation between physical activity and ethics, in which students can develop their ethics through physical education activities as students are greatly impacted by PE teachers. Jones suggested that PE teachers can play a model role, as displayed in physical education activities, in promoting their senses of virtue [7]. It is, therefore, that students’ ethics development would be, to some extent, dependent on PE teachers. In addition, in the relation with students and PE teachers, the latter ones are superior to the former ones, and in turn can determine students’ subject grades and coach sports teams, for example. Some teachers with poor ethics would probably treat students unfairly [8]. In this regard, it is encouraged that developing PE teachers’ ethics is beneficial to further promote students’ overall growth. Last, for effectiveness of physical education teaching, PE teachers with better ethics aim to develop students’ overall capability instead of only teaching how to move. This would further make those PE teachers positively explore more effective teaching methods in order to promote students’ development and health literacy, and finally improve teaching effectiveness.

Due to the importance of PE teacher ethics mentioned above, it has attracted much research interest. Some research has indicated that the teaching profession and teaching PE, in particular, are prone to abuse. For example, Mabagala demonstrated that anecdotal allegations concerning sexual harassment and abuse incidences during and after school sports and unfair play behavior during intramural and interschool competitions involving PE teachers are rampant in Tanzania [9]. Bennel also demonstrated that there was an increase in ethics abuses, such as sexual harassment and abuse in interschool sports competitions in Sub-Saharan Africa [10]. Based on the frequent incidents of ethical misconduct by PE teachers, Mabagala surveyed students’ perceptions of whether PE teachers comply with the Professional Code of Ethics and Conduct (PCEC) in Tanzania [11]. The results demonstrated that their teachers have a high level of compliance with the PCEC, which is contrary to the fact that PE teachers are abusive during interschool sports competitions and behave unfairly during intramural and interschool competitions. The contradiction between these research results and the reality suggests that PCEC cannot accurately assess PE teachers’ ethics. Consequently, an assessment specifically designed for PE teachers for their professional development is necessary. However, there is very little research on assessing the ethics of PE teachers. 

In ancient times, Confucius set high demands and standards for the ethics of Chinese teachers and proposed that teachers’ ethics should include attributes such as ‘Never be tired to educate your students’, ‘Be a model to others’, and ‘Be responsible, honest, and polite’. With the influence of traditional Confucian culture, the Chinese government has repeatedly emphasized the importance of teachers’ ethics in pedagogic development [12,13,14]. However, the current policies mainly focus on teachers and lack policies and action for the ethics of PE teachers. Over the past few years, incidents of PE teachers’ ethical misconduct have been made public more frequently, negatively impacting the noble image of teachers and demonstrating a reduction in the quality of PE. Therefore, it is necessary and urgent to develop an assessment of the ethics of PE teachers (AECPET). The aim of this study is to present the development of the AECPET theoretical model, identify the weights of AECPET’s domains and their indicators for assessment.

## 2. Materials and Methods

The successful development of the AECPET included two stages of research. In the first stage, the domains and indicators were identified using the Delphi method. The Delphi method was developed by the Rand Company in the early 1950s and was introduced in 1975. It is widely used to obtain opinions and suggestions from expert groups with relevant professional backgrounds [15]. In the second stage, the weight of each domain and indicator was identified using an expert ranking method. A web-based questionnaire survey was used to collect data in the two stages. The participants were invited to complete an online survey that was administered via a free online Chinese survey platform (https://www.wjx.cn, accessed on 25 August 2022).

### 2.1. Select and Qualify the Experts

To ensure the experts can provide credible and valid data for the research, a combination of objective and subjective methods was used to select and qualify experts for both the Delphi and expert ranking methods. In Section 2.1.1 Selecting Experts, the three selection criterions ensure that we can select experts objectively, and the expert familiarity criteria and expert judgment criteria subjectively test the qualifications of the experts themselves. 

#### 2.1.1. Selecting Experts

To ensure the progress of selecting experts is objective, we set three selection criterions as follows.


*Criterion 1: The experts should consist of PE teachers and researchers in the field of physical education teacher education (PETE) and ethic education.*


An expert is denoted as any one of the subjects we surveyed, including many professions such as teachers, researchers, officials, and leaders, among others. The AECPET is a theoretical model which aims to be used in practice. It is therefore important that the respondents should consist of both PE teachers with experience in teaching practice and researchers with experience in the theoretical research of PETE and ethic education.


*Criterion 2: The researchers should have published at least one paper on PETE or ethic education and the eligible PE teachers should have more than 5 years of teaching experience.*


Twelve researchers who had published at least one paper on PETE or ethic education and 13 PE teachers with teaching experience of over 5 years were selected through the Delphi method and expert ranking method. Across the 12 researchers, there was also an advanced PE teaching instructor who was a PETE researcher.


*Criterion 3: PE teachers from high, middle, and primary schools should be involved.*


As the current study aims to develop a theoretical model to assess the ethics of PE teachers who work in high, middle, or primary schools, it is necessary to include high, middle, and primary school PE teachers as study respondents. As such, the AECPET can be applicable for PE teachers from different layers of the Chinese schooling system. In the current study, three high school PE teachers, two middle school PE teachers, and two primary school PE teachers were included for survey.

Finally, 25 experts were sent questionnaire, and 15 of them responded in the first round of the Delphi survey. Then, in the second round of the Delphi survey, questionnaires were sent to the 15 experts, and all of them responded. In the following expert ranking method, we also received the opinions from these 15 experts. Last, the results from questionnaires filled by the 15 experts were used in this research (See in Figure 1).

Notes: UT means university teachers; HT means high school teachers; MT means middle school teachers; PT means primary school teachers; TS means teaching-research staff who have both researching and teaching experience; PETE means physical education teacher education; EE means ethic education.

#### 2.1.2. Qualifying Experts

For this research, 25 experts were invited. Finally, 15 experts agreed to be involved (See Table 1). To ensure that all the experts were qualified to participate in the research, the coefficients of expert familiarity and judgment were tested with the expert familiarity criteria and expert judgment criteria: (1)the expert should match their familiarity and judgment score with each domain; (2) the mean score of familiarity and judgment of eight domains was calculated; and (3) the mean score of expert familiarity and judgment was compared with 0.7 to indicate whether the expert is qualified [16] (see Table 2 and Table 3). All the scores for expert familiarity and judgment met the demands of the Delphi and expert ranking methods (mean score > 0.7), which means that the chosen experts were qualified to participate in this research. 

### 2.2. The First Stage: Delphi Method

First, from the perspective of government documents, laws, and the research of PE teachers [11,12,13], this study proposed 8 domains for the AECPET: (1) Policy Implementation (PI), (2) Legal Compliance and Patriotism (LCP), (3) Love for Students (LS), (4) Daily Performance (DP), (5) Philosophy of Educating Students (PES), (6) Attitude towards Scientific Research (ASR), (7) Awareness of Self-discipline and Honesty (ASH), and (8) Attitude towards Serving Society (ASS). Then, 38 indicators were defined according to the research of the professional development of PE teachers [17,18,19].

Second, the Delphi method, which is a process used to arrive at a group opinion or decision by surveying experts, was used to measure the objectivity and impartiality of the AECPET domains and indicators. Round iteration is the key point of the Delphi method, which can be used to identify the differences between experts’ opinions in the process of reaching a consensus. The results will not be derived in several survey rounds until the experts reach a consensus.

### 2.3. The Second Stage: Expert Ranking Method

In the second stage, the importance of the domains and indicators was sorted by the 15 experts, from which the weights of the domains and indicators were obtained. The expert ranking method is calculated using the following equation:(1)$$A_{j}=2[M\left(1+N\right)-R_{j}]/MN\left(1+N\right),$$
where *M* refers to the number of experts surveyed, *N* refers to the sum of the number of all indicators in the same rank, and *R* represents the rank-sum of the *j*th indicator. The rank-sum is the sum of the serial numbers of *M* experts for a certain indicator, which is represented by the letter *R*.

Then, according to the answers given by the experts, the domain weights were calculated. Second, according to the classification of the domains, the weights of the indicators were calculated, respectively. For example, there are three indicators for PL, and the total weight of the three indicators equals the weight of PL. Finally, the weight of the indicators was obtained by multiplying the weight of PL and the proportion of the indicators. If the weight of PL is 0.181, the proportions of the three indicators in PL are 0.444, 0.233, and 0.322 (the sum of the three is 1), and the proportions of the above three indicators in the AECPET are 0.181 × 0.444 = 0.081, 0.181 × 0.233 = 0.042, and 0.181 × 0.322 = 0.058. In summary, the several rounds of the Delphi process and expert ranking method are illustrated in Figure 2.

## 3. Results

### 3.1. Stage One: Identifying the Domains and Indicators

#### 3.1.1. The First Round of Delphi

By sending questionnaires to experts and inviting them to score the indicators anonymously, the following results were obtained (see Appendix A). If the average score > 3.0, standard deviation < 1, and coefficient of variation < 0.2, this means that the index has high credibility and should be retained [20]. The average value, standard deviation, and coefficient of variation of the 8 domains and 38 indicators were calculated following this criterion. In addition, the expert coordination coefficient, W, was 0.118 (*p* = 0.003 < 0.01), and the value of the coordination coefficient was between 0 and 1 [21]. The larger the coefficient, the higher the coordination of experts. However, the coordination was not high, which indicates that the experts had not reached a high degree of agreement, and that another round of Delphi should be carried out.

Some experts reflected that not fully implementing the guidance of sports work in the Chinese government’s policies should be added to PI, students not being guided to establish the concept of lifelong sports should be added to LS, unhealthy body shape due to bad living habits should be added to DP, and one-sided evaluation of students’ PE learning with a single evaluation method should be added in the PES. We accepted the suggestions and continued the second round of the Delphi method.

#### 3.1.2. The Second Round of Delphi

In accordance with the first-round results, four new indicators were added to the original framework, and the second round of the Delphi survey was conducted. The results are illustrated in Appendix A. All the results are in line with “average score > 3.0, standard deviation < 1, coefficient of variation < 0.2”, which means that all experts had a recognized attitude towards all indicators, and the final framework was determined. By doing so, the value of the coordination coefficient was improved to 0.496 (*p* = 0.000 < 0.01). Compared with the first round, this round had a better degree of coordination and consistency in the evaluation.

Finally, through two rounds of the Delphi survey, the overall framework of the evaluation criteria of the AECPET was developed (See Appendix B), consisting of 8 domains and 42 indicators. The number of indicators is moderate, and the language expression is clear and was highly recognized by experts.

### 3.2. Stage Two: Determining the Weight of Each Indicator

The expert ranking method was then used to determine the weight of each indicator in the overall framework of the evaluation criteria of the AECPET. According to the weight of each index, the AECPET index system was developed. In this round of expert ranking, to ensure the consistency of the answers given by the experts, it was necessary to import questionnaire data into SPSS online analysis software (SPSS Inc., Chicago, IL, USA) to test the consistency of the expert opinions. The Kendall W coefficient was 0.793 (*p* = 0.000 < 0.01), which indicates that all the experts reached a consensus in selecting the indicators from the questionnaire.

Finally, through the two Delphi rounds and one expert ranking round, we drew up the AECPET index system (See Table 4). The domains weight order is as follows: LCP (0.191) > PI (0.181) > PES (0.163) > LS (0.162) >DP (0.111) > ASH (0.091) > ASR (0.072) > ASS (0.046).

## 4. Discussion

The AECPET can help provide an overall understanding of teachers’ ethics in China under the influence of Confucianism. It can also be used to understand the differences between Chinese PE teachers’ ethics and other teachers’ ethics. The special ethics of PE teachers have a large impact on PE and health education, and we expect that the AECPET index system will promote PE and health education.

The results demonstrate the ethics that Chinese PE teachers should have and the ranking of the importance of ethics. Through the ranking of domains, it is evident that the weight of the AECPET index system has three levels: (1) LCP and PI; (2) PES, LS, DP and ASH; (3) ASR and ASS.

LCP and PI are the basic components of the literacy of PE teachers, in the first place of AECPET, as the development orientation of education is guided by the Chinese government. PE teachers are the implementors of policies published by the government. As a result, the level of LCP and PI, to a certain extent, can affect the effectiveness of policy [22]. The Physical Education and Health Curriculum Standards for Compulsory Education (2022 Edition) proposes that PE teachers should play a role in conducting health education [23]. PE teachers with better LCP and PI would adhere to government policies to help build the Chinese national public health system. Among other countries, PE teachers should undertake responsibilities with the guidance from governments. PE teachers in the USA, for example, are required to help students foster personal values that support health; develop group norms that value a healthy lifestyle; and develop the essential skills necessary to adopt, practice, and maintain health-enhancing behaviors [24]. In some countries, PE teachers need to fulfill the duty and responsibilities set by the government; as a result, LCP and PI are the most important part in the AECPET.

The weights of domains of PES, LS, PD, and ASH rank from 3rd to 6th. They are related to the performance of PE teachers’ teaching practice. In the 21st century, PE teachers are supposed not only to instruct students on how to move but also to promote students’ health and well-rounded development [25]. Therefore, it is important to clarify the goal within the AECPET [26]. Within the domain of PES, most countries around the world aim to develop the core competencies of students. In physical education activities, PE teachers have the option to either only conduct movement skills teaching or to design the teaching progress according to students’ distinctive personalities and promote physical literacy. Their different choices may have lifetime effects on students’ lifestyle habits. The former choice can only develop students’ motor skills, while the latter would help students learn how to participate in sports, and it promotes health and raises virtues. Within the domain of LS, as PE lessons are based on physical activity, to prevent psychological harms and physical injuries, PE teachers should pay attention to the physiological and psychological changes of students. In some other countries, LS is an important component of the PE teachers’ professional code of ethics; for example, the code of ethics for PE teachers in Turkey requires that teachers should care for the physical and psychological well-being of students [27]. In the dimension of DP, teachers are the representatives of morality in Chinese Confucian culture, but PE teachers have long been misunderstood as “physically strong but simply minded and rude”. Therefore, PE teachers should avoid rude words and behaviors in their daily lives and work in order to improve the image of PE teachers in people’s minds. In other cultures, PE teachers who behave rudely and use profanity would frequently damage their personal images and insult the profession. For the students, the education of ethical behavior in PE is one of the educational goals; therefore, teachers should be a model for students in daily life [28]. Therefore, a high level of DP is also crucial for PE teachers to implement ethical education as a moral role model. Within the dimension of ASH, in China, PE teachers are authorized to enroll students in sports teams. To avoid PE teachers taking bribes in enrollment and helping students cheat to get into school or become the mainstay of the team, ASH is also an important part in the AECPET. In other countries, ASH is the virtues that PE teachers must rise to, and it is also necessary to avoid PE teachers helping student to cheat in the PE exam. For example, the PE teachers’ code of ethics and conduct in Tanzania [11] and the USA [29] both include honesty and integrity as an essential part of the PE teacher’s professional ethics.

The weights of the domains of ASR and ASS, representing PE teachers making contributions to society and the academic sector by practical experience, ranked from 7th to 8th. In the domain of ASR, because PE teachers are experienced in teaching practice, they can obtain research output from teaching through observing students’ performance, collecting data from class, and conducting teaching experiments. As the PE curriculum reform goes forward, relevant research on the PE curriculum should keep pace with the reforms. It is highly valuable that PE teachers can provide advice and research outcomes from practical experience to promote the PE curriculum. To better advance PE teachers’ research, promoting ethnics is also urgently needed [30]. Within the domain of ASS, PE teachers’ participation in social services and disseminating health and exercise knowledge is one of the approaches for PE teachers to achieve their personal development. In the context of “Healthy China 2030”, people’s pursuits of health have received much attention [31]. Owing to academic and work pressure, people sit in classrooms and offices for longer durations, which negatively affects their health [32]. In addition to teaching and education, PE teachers with good ethics can actively participate in public activities such as fitness instruction and health disseminations.

In summary, the AECPET is an evaluating system constructed based on the perspective of Chinese PE teachers, which also has a potentially global perspective and can be used by PE teachers with different cultural views.

The results of this study can be used for developing PE teachers in various ways. First, at the government level, improvements should be made to the standard educational framework of PE teacher training [33], and to the content of teachers’ morals and assessment. Second, the AECPET should be the standard when qualifying excellent teachers. A zero-tolerance attitude towards the AECPET should be developed to reduce the probability of ethics problems. If a PE teacher makes a mistake that was included in the AECPET, they will be punished according to the degree of the mistake. In addition, salary incentives should be introduced for teachers with high ethics levels, the economic needs of PE teachers should be met [34], and the level of PE teachers’ ethics should be increased through active publicity and positive guidance.

At the school level, we should pay attention to the diversified characteristics of the AECPET, which include the teacher’s invisible morality, dominant behavior and common index of ordinary teachers, and the characteristic index of PE teachers. Therefore, as the first standard to evaluate PE teachers, we should consider all domains when using the AECPET and pay attention to the impact of the AECPET characteristics on teachers’ common ethics index. Moreover, in the evaluation of the AECPET, schools should avoid putting forward requirements that are too high, and gradually improve the PE teachers’ level of ethics [35].

At the level of PE teachers, subjective initiatives should be used to consciously improve self-cultivation, teachers’ moral levels, and teaching practice norms. Teachers also need to understand the scope of the AECPET, actively participate in political learning activities, improve their ideological understanding, and reflect by comparing themselves to the evaluation indicators of the AECPET. PE teachers can compare their thoughts and behaviors with the AECPET index system to consider anomalies or trends in each index dimension. They can also reflect deeply, change, encourage, and improve the level of ethics through self-cultivation. Additionally, Yin’s research demonstrates that we should pay more attention to teachers’ sense of social responsibility when confronting major public crises [36]. Subsequently, combined with the reality of the health crisis caused by the COVID-19 pandemic, teachers with positive ethics can help manage major epidemics and form correct guidance, which is vital for social stability and social harmony [37].

At the student level, we should focus on their supervision power, always remind teachers to regulate their behavior, and help PE teachers improve their ethics in teaching practice through teacher–student interactions. As a group in direct contact with PE teachers, students are often victims of teachers’ bad ethics, such as being forced to attend “sheep classes” (teach nothing and let students do what they want to), being physically punished, and being physically assaulted by PE teachers. Therefore, students should use the AECPET to assess PE teachers and provide student supervision to develop PE teachers.

Although the AECPET can provide an assessment tool for PE teachers, there are still some limitations. First, the number of involved experts is small, which may weaken the credibility of the results. In the future, the research should involve more experts (1) in more locations, (2) covering all age of experts, (3) from different countries, etc. Second, the descriptions of indicators are verbalized negatively. In China, there is no assessment especially designed for the ethics of PE teachers. We should let PE teachers know what the ethical baseline is, along with the things they should not do. At the present stage, we mainly assessed whether PE teachers do things that violate professional ethics. When the level of ethics improves, we will change the descriptions into positive forms to let PE teachers know what “Ethical Delegates” should do. Third, The AECPET is developed for Chinese PE teachers; it can provide an example for other countries but cannot be directly applied to the evaluation of PE teachers in other countries. Based on AECPET, researchers in other countries should consider some specific indicators adapted from their culture when developing professional ethics assessments for PE teachers.

## 5. Conclusions

The Assessment of Ethics for Chinese PE teachers (AECPET) is a theoretical model and a tool that can be used to support the assessment of PE teachers in practice. It is the first research in China to construct an evaluating system that can help Chinese and international researchers better understand PE teachers’ ethics. The AECPET includes eight domains and 42 indicators. The weight of each domain and indicator can reflects the degree of importance. The government, schools, PE teachers, and students can apply AECPET to assess the ethics of PE teachers and help PE teacher improve their level of ethics.

## Figures and Tables

**Figure 1 ijerph-19-11905-f001:**
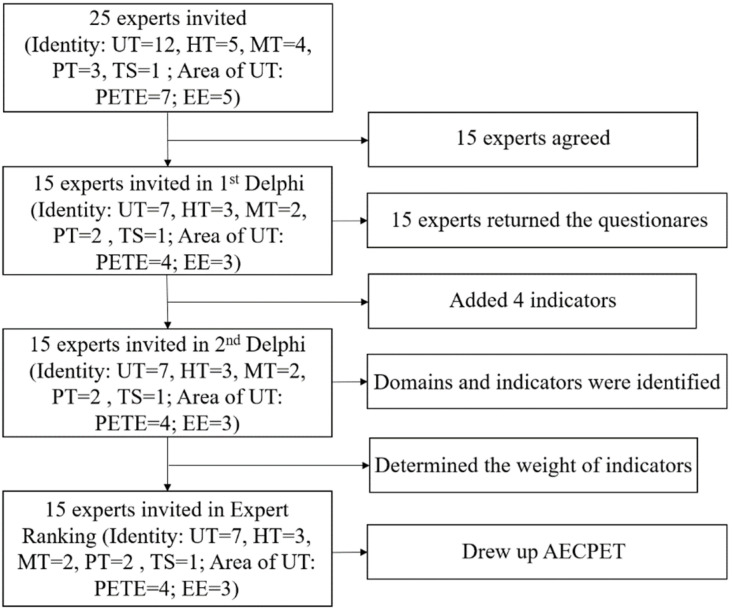
Procedure of the Delphi and expert ranking process.

**Figure 2 ijerph-19-11905-f002:**
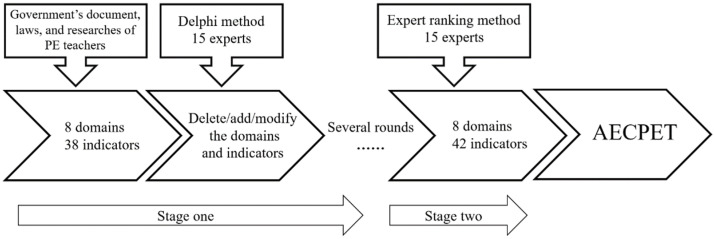
The AECPET research design.

**Table 1 ijerph-19-11905-t001:** Basic Information of Experts.

Number	Age	Degree	Title	Identity
E01	>50	M.Ed.	Senior teacher ^1^	Teaching-research staff
E02	30–39	Ph.D.	Associate professor	University teacher
E03	30–39	Ph.D.	Associate professor	University teacher
E04	40–49	Ph.D.	Associate professor	University teacher
E05	40–49	M.Ed.	Associate professor	University teacher
E06	30–39	Ph.D.	Associate professor	University teacher
E07	30–39	M.Ed.	Assistant professor	University teacher
E08	30–39	M.Ed.	Assistant professor	University teacher
E09	40–49	B.Ed.	Senior teacher	High school teacher
E10	>50	B.Ed.	Senior teacher	High school teacher
E11	40–49	B.Ed.	Senior teacher	High school teacher
E12	40–49	B.Ed.	Senior teacher	Middle school teacher
E13	>50	B.Ed.	Senior teacher	Middle school teacher
E14	30–39	B.Ed.	First-class teacher ^2^	Primary school teacher
E15	30–39	B.Ed.	Second-class teacher ^3^	Primary school teacher

Notes: ^1^ A senior teacher is the equivalent of an associate professor; ^2^ A first-class teacher is the equivalent of an assistant professor; ^3^ A second-class teacher is the equivalent of a teaching assistant.

**Table 2 ijerph-19-11905-t002:** Expert Familiarity Criteria.

Degree	Score
Very familiar	1
Relatively familiar	0.8
Fair	0.6
Relatively unfamiliar	0.4
Very unfamiliar	0.2

**Table 3 ijerph-19-11905-t003:** Expert Judgment Criteria.

Judgment Basis	Score
Theoretical analysis	1
Practical experience	0.75
Peer understanding	0.5
Intuitive perception	0.25

**Table 4 ijerph-19-11905-t004:** The AECPET index system.

Domains	Indicators
Policy Implementation (0.181)	Violating the core values of the culture principles of Chinese and the Chinese government’s policies. (0.081)
	Not fully implementing the guidance of sports work in the Chinese government’s policies. (0.042)
	Maliciously slandering the Chinese government and violating the Chinese government’s guiding policies. (0.058)
Legal compliance and patriotism (0.191)	Colluding with foreign forces to obtain information about Chinese School PE (0.061)
	Slander against China when talking about international sports (0.042)
	Failing to perform the basic duties of PE teachers by national laws and regulations (0.048)
	Violation of various sports laws and regulations issued by the state (0.039)
Love for students (0.162)	Not paying attention to students’ physiological and psychological states in PE class (0.026)
	Students are required to complete many exercise tasks beyond their ability level (0.021)
	Physical punishment of students by overloading exercise (0.024)
	Abusive criticism when there are mistakes and failures in the students’ movement (0.022)
	Ignore the needs of special students (0.017)
	In the classroom teaching, only complete the task of PE without considering the actual situation of students (0.017)
	Students are not guided to establish the concept of lifelong sports (0.018)
Daily Performance (0.111)	Swearing in and out of PE class and daily life (0.018)
	Clothing not consistent with the health image of PE teachers (0.017)
	Improper relationship with students or obscene and sexual harassment of students (0.029)
	Operating sideline work outside school has a negative impact on school PE (0.016)
	Comments that belittle the profession of PE teachers (0.020)
	Unhealthy body shape due to bad living habits (0.011)
Philosophy of educating students (0.163)	Ignoring the basic tasks and requirements of “cultivating morality through sports” (0.030)
	The phenomenon of “herding sheep” in PE classroom teaching (0.021)
	Narrowing PE into pure competitive sports training (0.017)
	Despise the cultivation of students’ sports morality in PE (0.024)
	Neglecting the cultivation of students’ healthy behaviors in PE (0.022)
	Take a perfunctory and coping attitude to school PE (0.018)
	Pass on concepts contrary to the core values to students in school PE (0.023)
	One-sided evaluation of students’ PE learning with a single evaluation method (0.008)
Attitude towards scientific research (0.072)	Lack of proper cognition of sports science research (0.016)
	Plagiarism of sports research papers or falsification of academic ethics (0.019)
	Usurping the sports research achievements of students, colleagues, and others (0.016)
	Using their influence in sports academic circles to trade interests with others (0.013)
	Indifference to sports research (0.007)
Awareness of self-discipline and honesty (0.091)	Accepting different forms of bribery in sports enrollment, training, and competition (0.027)
	Modifying the results of PE or sports competition for students by improper means (0.021)
	Seeking convenience through fraud, bribery, and other means in the work of sports excellence, prize, and first prize evaluations (0.020)
	Selling sportswear and equipment to students or parents for profit (0.130)
	Accept banquets, tourism, entertainment, and leisure activities paid for by students or parents (0.011)
Attitude towards serving society (0.046)	Unauthorized use of schools’ names or other information to set up sports training classes (0.010)
	Refuse to participate in sports public welfare activities arranged by superior departments or schools (0.009)
	Abuse of influence in sports in society (0.014)
	Using the students’ sports expertise to seek personal interests in society (0.013)

## Data Availability

Data can be shared upon reasonable request to the corresponding author.

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
