# Peer review of "Development of an Assessment of Ethics for Chinese Physical Education Teachers: A Study Using the Delphi and Expert Ranking Methods"

_ijerph, 2022, doi:10.3390/ijerph191911905_

Round 1

Reviewer 1 Report (Previous Reviewer 2)

The quality of the corrected manuscript has been significantly improved compared with the raw manuscript. Developing an ethics assessment for PE teachers in China is indeed helpful in improving the quality of the PE teachers in China. 

There are just two pieces of advice for authors as follows:

1. The description of Experts’ Selection is confusing. Three concepts used in the description: expert, researcher and PE teacher, but without distinguishing.

2. The credibility of the results may be weakened based on the small number of involved experts.

Author Response

Reviewer 2 Report (New Reviewer)

Thank you very much for the opportunity to review this article. Below I express some concerns and suggestions to improve the quality of the article.

First of all, I don't think the title of the paper is appropriate.

"An Assessment of Ethics for Chinese Physical Education Teachers: A Study Using the Delphi and Expert Ranking Methods"

The authors do not evaluate the ethics of Chinese Physical Education, what authors did was an evaluation tool. In my opinion, you should modify the title to adjust it to the objective of the paper. I think that more appropriate would be “Development of a tool for the evaluation…

Some comments on the content of the paper

Line 43. I think that PE is the first time it appears in the manuscript (except the abstract), so it would be necessary to include the abbreviation again -eg Physical Education (PE) teachers –

Line 108. I am confused, if the experts were selected by the Delphi method, what initial criteria were established to reach the final criteria of 5 years of experience, and could they better clarify the "expert ranking method". Is the same of Expert Familiarity Criteria?

Line 131. Add “25 experts were invited. Finaly 15 experts agree…”

Line 136 and 137. Probably the authors can add (see table 2 and see table 3) in this paragraph.

Line 140. –This comment is not mandatory, it is only a suggestion or reflection- I have notice that all the participants are over 30 years old. Probably motivated by a inclusion criterion of 5 years of experience, but I think that the authors should point out as a limitation of the study that they did not include young people who might meet the expert criteria (for example, a young researcher who has published an article about ethics)

Line 194. “Some experts reflected that Not fully” change Not for not

This article has numerous limitations that should be pointed out and should be written at the end of the discussion and before the conclusions. Several limitations occur to me, such as the concept of Eastern ethics may be different from the concept of Western ethics, and perhaps some constructs cannot be compared. China is a very big country, I don't know if the experts are from a specific geographical area or if they represent different parts of China, and another limitation is the number of experts, for example, there are only two or three per educational level. I think that the authors should explain all the limitations they have found in their work for a better understanding.

Author Response

Reviewer 3 Report (New Reviewer)

The study aimed to develop an assessment of ethics for Chinese PE teachers. Even if the authors described their understanding of teachers ‘ethics (p. 1, l. 35-36), a differentiation from bioethics seems important in the area of PE. Furthermore, it is not clear, if the authors want to assess more the possibilities of physical activity to improve morality itself in the context of PE or the attitude, knowledge and behaviour of PE teachers. My suggestion is to clearly describe the intention of the approach.

In the introduction part some assertions are not supported by literature: “Last, for effectiveness pf physical education teaching, PE teachers with better ethics aim to develop students ‘overall capability instead of teaching how to move (p. 2, l. 53-55). And, it is necessary to explain the relationship between physical activity and ethics (p. 2, l. 44).  In the introduction but also in the discussion the connection of ethics and health literacy and physical literacy respectively, is unclear.     

The stages of the developed AECPET are described in such a way that the reader is sometimes unable to follow the individual steps. In particular, it is noticeable that the indicators in table 4 are expressed positively while the descriptions are verbalized negatively, is this the intention of the authors?

In the discussion part connections of the new assessment with health education are described and perspectives are derived from it: “(…) we expect that the AECPET index system will promote PE and health education” (p. 8, l. 230-231). Unfortunately, this argumentation is hardly comprehensible for the reader, since no justifications are provided.   

Round 2

Reviewer 3 Report (New Reviewer)

Dear authors,

thanks for the revision, in some places the objective and approach is now much clearer. However, the transferability to PE in general is not clear from my point of view and the connection with physical literacy and health literacy is hardly comprehensible, especially with regard to Whithead's concept. The question is if these aspects need to be linked to the topic of PE. teachers´ethics at all.

This manuscript is a resubmission of an earlier submission. The following is a list of the peer review reports and author responses from that submission.

Round 1

Reviewer 1 Report

Line 12. It is no vital, it isimportant. Change the word.

The text has too many phrases and expressions in quotation marks. I recommend sparing the use of phrases and expressions in quotes.

Lines 54-55. The rationale for developing ethics for teachers is important. However, the justification cannot be based on the fact that teachers from other countries have codes of ethics.

Line 56. "Due to the importance of PE teachers' ethics". What exactly is the importance of PE teachers’ ethics? If this is important, it must be well expressed in the text.

At the end of the introduction, it is not clear what the purpose of the article is.

The methodology is too vague. How were teachers selected to enter the Delphi panel?

This information is very important.

Line 141. How were the questionnaires sent?

My question may seem strange, but for the development of a code of ethics is there a political domain related to the values ​​of a political party?

This seems contrary to ethical values because ethics should have nothing to do with each person's political ideology.

If the code of ethics is held captive by a political party, then it is a code of subservience by a political party.

The article's content is highly politicized. Science must not be politicized to be neutral and impartial. Whenever science exalts a political leader, a party or an ideology, it risks being biased.

Author Response

Response to Reviewer 1: Thank you for your review of our paper. We have answered each of your points below.

Q1: Line 12. It is no vital, it is important. Change the word.

A2: Thanks for the reviewer’s suggestion, we have revised it, please see Line 12.

Q2: The text has too many phrases and expressions in quotation marks. I recommend sparing the use of phrases and expressions in quotes.

A2: Thank you very much and we have revised them accordingly, please see the revised manuscript we uploaded.

Q3: Lines 54-55. The rationale for developing ethics for teachers is important. However, the justification cannot be based on the fact that teachers from other countries have codes of ethics.

A3: Thanks for the reviewer’s comment. We agree the fact that other countries have stressed ethical standards specifically for physical education (PE) teachers, which cannot be ruled out and China should also stressed ethical standards specifically for PE teachers. For this reason, this paragraph has been deleted from this the revised manuscript and describe the importance of ethics of PE teachers instead. Please see Line 43-57.

Q4: Line 56. "Due to the importance of PE teachers' ethics". What exactly is the importance of PE teachers’ ethics? If this is important, it must be well expressed in the text.

A4: Thank you very much! In this round of revision, we focus on the importance of ethics for PE teachers in the second paragraph:

PE teacher ethics is of great importance. In terms of students’ ethics development, Drewe indicated that there was a relation between physical activity and ethics, which students can develop their ethics through physical education activities as students are greatly impacted by PE teachers. Jones suggested that PE teachers can play a model role, as displayed in physical education activities, in promoting their senses of virtue. It is therefore that students’ ethics development would be, to some extent, dependent on PE teachers. In addition, in the relation with students and PE teachers, the latter ones are superior to the formers, which in turn can determine students’ grades of subject and coach sports team, for example. Some teachers with poor ethics would probably treat students unfairly. In this regard, it is encouraged that developing PE teachers’ ethics is beneficial to further promote students’ overall growth. Last, for effectiveness of physical education teaching, PE teachers with better ethics aim to develop students’ overall capability instead of teaching how to move. This would further make those PE teachers positively explore more effective teaching methods, in order to promote students’ development, health literacy and finally improve the teaching effectiveness.

‘Due to the importance of PE teacher ethics above’ is used in the third paragraph to connect with the specific connotation of the importance of PE teachers' ethics in the second paragraph.

Please see Line in 43-57 in the revised manuscript.

Q5: At the end of the introduction, it is not clear what the purpose of the article is.

A5: Thank you very much. This revision further specifies the purpose of this study: The aim of this study is to present the development of the AECPET theoretical model, identify the weights of AECPET’s domains and their indicators for assessment.

Please see Line in 84-86 in the revised manuscript.

Q6: The methodology is too vague. How were teachers selected to enter the Delphi panel? This information is very important.

A6: Thank you very much. In this revision, we add the criteria for selection of teachers:

Criterion 1: The experts should consist of PE teachers and researchers in the field of physical education teacher education (PETE) and ethic education.

The AECPET is a theoretical model which aims to be used in the practice. It is therefore that the respondents should both consist of PE teachers with experience in teaching practice and researchers with experience in the theoretical research of PETE and ethic education.

Criterion 2: The researchers should have published at least one paper on PETE or ethic education and the eligible PE teachers should have more than 5 years for teaching experience.

12 researchers who have published at least one paper on PETE or ethic education and 13 PE teachers with teaching experience over 5 years were selected through Delphi method and expert ranking method. Across the 12 researchers, there was an advanced PE teaching instructor who was a PETE researcher.

Criterion 3: PE teachers from high, middle and primary schools should be involved.

As the current study aims to develop a theoretical model to assess the ethics of PE teachers who works in high, middle or primary schools. It is necessary to include high, middle and primary schools PE teachers as study respondents. As such, the AECPET can be applicable for PE teachers from different layers in Chinese schooling system. In the current study, three high school PE teachers, two middle school PE teachers and two primary school PE teachers were included for survey.

Finally, 25 experts were questionnaires sent questionnaire while 15 of them responded in the first round of Delphi survey. Then, in the second round of Delphi survey, the questionnaires were sent to the 15 experts and all of them responded. In the following expert ranking method, we also received the opinions from these 15 experts. Last, the result of questionnaires filled by 15 experts were used in this research.

Q7: Line 141. How were the questionnaires sent?

A7: Thanks for the questions from the reviewer. A web-based questionnaire survey was used to collect data in the two stages. The participants were invited to complete an online survey that was administered via a free online Chinese survey platform (https://www.wjx.cn)

Q8: My question may seem strange, but for the development of a code of ethics is there a political domain related to the values of a political party?

This seems contrary to ethical values because ethics should have nothing to do with each person's political ideology.If the code of ethics is held captive by a political party, then it is a code of subservience by a political party.

The article's content is highly politicized. Science must not be politicized to be neutral and impartial. Whenever science exalts a political leader, a party or an ideology, it risks being biased.

A8: Thanks to the reviewer's question, the Political Literacy in the original article is not clearly formulated, and in scientific research, the research results should be distinguished from party and government beliefs. From a global perspective, ethics really have nothing to do with each person's political ideology, and the reference to Political Literacy in this paper is in fact the support and implementation of government policies by PE teachers.

For this reason, this revision changed the term Political Literacy to Policy Implementation: â‘ Violating the core values of socialism with Chinese characteristics and the Communist Party of China's (CPC) political policy and basic line was changed to Violating the core values of culture principles of Chinese and the Chinese government’s policies. â‘¡Not fully implementing the guiding spirit of sports work in the CPC's major meetings and documents was amended to Not fully implementing the guidance of sports work in the Chinese government’s policies. â‘¢The Chinese government's governance instruments should be changed from Maliciously slandering the CPC and violating the CPC's Political Thought to Maliciously slandering the Chinese government and violating the Chinese government’s guiding policies.

Reviewer 2 Report

Issues need to be answered or improved by the authors:

1. What are the criteria for experts? Are the methods (table 2 & table 3) used in the study to assess the quality of experts subjective? If it is subjective, how do we ensure the credibility and validity of the research results?

2. Why are some of the experts participating in the study from primary or secondary schools? (Table 1) . I think there is a big difference between primary and secondary school students. Secondary school teachers and primary school teachers should be explained separately. The experts from primary and secondary schools in this study are insufficient.

3. There is an item called political literacy in the AECPET obtained by the study involving a political party (the Communist Party of China). The political party background of all experts needs to be stated. Are all experts from CPC?

The reliability of the results is low due to the significant issues in selecting experts in this study.

Author Response

Response to Reviewer 2: Thank you for your comments. Our answers to your points are as follows

Q1: What are the criteria for experts? Are the methods (table 2 & table 3) used in the study to assess the quality of experts subjective? If it is subjective, how do we ensure the credibility and validity of the research results?

A1: Thank you very much. In this revision, we add the criteria for selection of teachers:

Criterion 1: The experts should consist of PE teachers and researchers in the field of physical education teacher education (PETE) and ethic education.

The AECPET is a theoretical model which aims to be used in the practice. It is therefore that the respondents should both consist of PE teachers with experience in teaching practice and researchers with experience in the theoretical research of PETE and ethic education.

Criterion 2: The researchers should have published at least one paper on PETE or ethic education and the eligible PE teachers should have more than 5 years for teaching experience.

12 researchers who have published at least one paper on PETE or ethic education and 13 PE teachers with teaching experience over 5 years were selected through Delphi method and expert ranking method. Across the 12 researchers, there was an advanced PE teaching instructor who was a PETE researcher.

Criterion 3: PE teachers from high, middle and primary schools should be involved.

As the current study aims to develop a theoretical model to assess the ethics of PE teachers who works in high, middle or primary schools. It is necessary to include high, middle and primary schools PE teachers as study respondents. As such, the AECPET can be applicable for PE teachers from different layers in Chinese schooling system. In the current study, three high school PE teachers, two middle school PE teachers and two primary school PE teachers were included for survey.

Finally, 25 experts were questionnaires sent questionnaire while 15 of them responded in the first round of Delphi survey. Then, in the second round of Delphi survey, the questionnaires were sent to the 15 experts and all of them responded. In the following expert ranking method, we also received the opinions from these 15 experts. Last, the result of questionnaires filled by 15 experts were used in this research.

Q2: Why are some of the experts participating in the study from primary or secondary schools? (Table 1). I think there is a big difference between primary and secondary school students. Secondary school teachers and primary school teachers should be explained separately. The experts from primary and secondary schools in this study are insufficient.

A2: We thank for the great idea from the reviewer. We agree that there is a big difference between primary and secondary school students. Since this study is aimed at developing assessment of ethics of PE teachers in K-12, not high school PE teachers, middle school PE teachers, or elementary school PE teachers. The framework of AECPET can cover K-12 PE teachers’ ethics. As to the specific ethics of PE teachers, it can be developed by the primary school teachers, secondary school teachers and their principals.

Q3: There is an item called political literacy in the AECPET obtained by the study involving a political party (the Communist Party of China). The political party background of all experts needs to be stated. Are all experts from CPC?

A3: Thanks to the reviewer's question, the Political Literacy in the original article is not clearly formulated, and in scientific research, the research results should be distinguished from party and government beliefs. From a global perspective, ethics really have nothing to do with each person's political ideology, and the reference to Political Literacy in this paper is in fact the support and implementation of government policies by PE teachers. For this reason, this revision changed the term Political Literacy to Policy Implementation: â‘ Violating the core values of socialism with Chinese characteristics and the Communist Party of China's (CPC) political policy and basic line was changed to Violating the core values of culture principles of Chinese and the Chinese government’s policies. â‘¡Not fully implementing the guiding spirit of sports work in the CPC's major meetings and documents was amended to Not fully implementing the guidance of sports work in the Chinese government’s policies. â‘¢The Chinese government's governance instruments should be changed from Maliciously slandering the CPC and violating the CPC's Political Thought to Maliciously slandering the Chinese government and violating the Chinese government’s guiding policies.

Q4: The reliability of the results is low due to the significant issues in selecting experts in this study.

A4: We thank for the reviewer’s comments. In our previous article, we did not describe the process of selecting experts in detail, so in this revision we have further detailed the specific steps of expert selection. First, a combination of objective screening of experts by researchers and subjective self-assessment of experts' qualifications ensures the authority of experts in the field and the reliability of feedback. Second, by consulting PE teachers and researchers in the field of ethics research to obtain authoritative information at the theoretical level, and by consulting PE teachers to obtain reliable opinions at the practical level, the combination of theory and practice ensures that the AECPET has both theoretical support and can be applied in practice. By including the above information in the text, the complete process of selecting experts can be showed to the reader and ensure the authority of the experts in order to improve the reliability of the research results.